# Students Parrot Their Teachers: Membership Inference on Model Distillation

**Matthew Jagielski**[1]     **Milad Nasr**[1]     **Katherine Lee**[1]
**Christopher Choquette-Choo**[1]     **Nicholas Carlini**[1]     **Florian Tramèr**[2]

[1]*Google DeepMind*
[2]*ETH Zurich*

## Abstract

Model distillation is frequently proposed as a technique to reduce the privacy leakage of machine learning. These empirical privacy defenses rely on the intuition that distilled "student" models protect the privacy of training data, as they only interact with this data indirectly through a "teacher" model. In this work, we design membership inference attacks to systematically study the privacy provided by knowledge distillation to both the teacher and student training sets. Our new attacks show that distillation alone provides only limited privacy across a number of domains. We explain the success of our attacks on distillation by showing that membership inference attacks on a private dataset can succeed even if the target model is *never* queried on any actual training points, but only on inputs whose predictions are highly influenced by training data. Finally, we show that our attacks are strongest when student and teacher sets are similar, or when the attacker can poison the teacher set.

## 1   Introduction

Model distillation [HVD+15] is a common framework for knowledge transfer, where knowledge learned by a "teacher model" is transferred to a "student model" via the teacher's predictions. Distillation is helpful because the teacher's predictions are a more useful guide for the student model than hard labels; this phenomenon has been explained by the teacher's predictions containing some useful "dark knowledge". Variants of model distillation have been proposed for, e.g., model compression [HVD+15; BC14; PPA18; KPK18; SCGL19] or training more accurate models [ZK16; XLHL20]. Within the privacy-preserving machine learning community, distillation has been adapted to protect the privacy of a training dataset [PAEGT16; TMSSNHM22; SH21; MHHVEHP22].

Many of these approaches rely on the intuition that distilling the teacher model serves as a privacy barrier that protects the teacher's training data. Informally, restricting the student to learn only from the teacher's predictions is a form of *data minimization*, which should result in less private information being fed into, and memorized by, the student. This privacy barrier around the teacher also allows the teacher model to be trained with strong, non-private, training approaches, improving both the teacher model's and student model's accuracy.

Because model distillation does not provide a rigorous privacy guarantee (such as those offered by differential privacy [DMNS06]), in our work we evaluate the empirical privacy provided by these schemes. We show that distillation is vulnerable to membership inference attacks—a well-studied class of privacy attacks on machine learning [SSSS17; YGFJ18].

We adapt the state-of-the-art Likelihood Ratio Attack (LiRA) [CCNSTT22] to the distillation setting, and find that this attack works surprisingly well at inferring membership of the teacher's training data. For some training examples, model distillation fails to appropriately protect against mem-

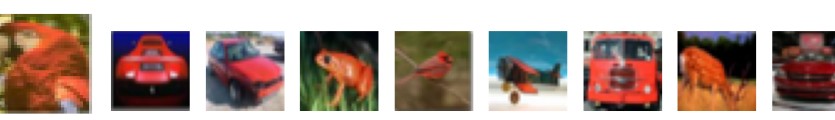

(a) Target                                    (b) Indirect Student Queries

Figure 1: We can predict the membership status of a target example (a) in the teacher model's training set by querying the teacher model on different student training examples (b). Our membership inference attack uses only the student model's predictions on the ten images on the right to reach 73% accuracy at predicting membership of the target image. Interestingly, while the target example is a bird, only two of the most informative student queries are birds.

bership inference. To explain this finding, we show for the first time that the membership presence of some examples can be inferred based on the model's predictions on *other*, seemingly unrelated examples. This observation provides new insights into how membership information is transmitted from the teacher to the student.

Figure 1 gives an example of such cross-example leakage. A teacher model trained on the red parrot in Figure 1a (labeled as "bird"), never seeing the student queries on the right in Figure 1b, encodes membership information about the parrot in predictions on these student queries. Interestingly, the most informative student queries are not birds—they are images of other classes with similar red hues that the model *confuses* as being "bird-like" because of the influence of the parrot in the teacher's training set. In other words, the student model manages to "parrot" its teacher using the peculiarities transferred through distillation.

We systematically evaluate a number of factors which impact the empirical privacy of model distillation. We find that similarity between the training datasets of the teacher and student models lead to increased privacy risk, along with higher temperature parameters. Moreover, an adversary capable of poisoning the teacher training set can amplify privacy risk for teacher examples, in line with recent work [TZJRR16; CSSWZ22]. We also find that distillation provides little privacy protection to student examples. We hope our attacks can assist experts in properly evaluating the privacy risks resulting from model distillation. Our work also highlights two possible approaches for mitigating privacy risks in distillation: deduplicating the teacher and student datasets, and reducing the leakage from the teacher model, e.g., using provable guarantees such as differential privacy [DMNS06].

## 2    Background and Related Work

### 2.1    Machine Learning Privacy

Machine learning models are known to be vulnerable to a variety of privacy attacks, including membership inference attacks [SSSS17], attribute inference attacks [FJR15], property inference attacks [GWYGB18], and training data extraction [CTWJHVLRBSE+21]. Each of these captures a different type of leakage about the training data or individual examples.

In this work, we focus on membership inference, as it is the most widely studied privacy attack. In a membership inference attack, an adversary tries to determine whether or not a particular example was used to train a model. There have been a number of membership inference attacks proposed in the literature, which generally compute some "membership score", which is designed to be informative of a target example's membership. Most of these scores make use of the model's prediction on the target example, perhaps to compute the example's loss [YGFJ18; SDSOJ19] or to compute some other score [WGCS21; SSSS17; CCNSTT22]. Other attacks rely on querying the model with examples nearby or derived from the target [JWKGE20; LBWBWTGC18; LZ21; CCTCP21; WBKBGGG22]. One of our contributions is to design an attack which performs well despite relying on the model's predictions on entirely different examples from the target.

Our attacks are based on the state-of-the-art Likelihood Ratio Attack (LiRA) [CCNSTT22]. In LiRA, the adversary first trains many *shadow models*, such that half of these models will contain a given target example $(x, y)$ (the IN models), and half will not (the OUT models). Next, the adversary queries each shadow model $f$, to compute the logits corresponding to the correct class, $f(x)_y$, and fits a Gaussian $\mathcal{N}(\mu_{\text{in}}, \sigma^2_{\text{in}})$ to the logits for all models containing the example (and similarly for the

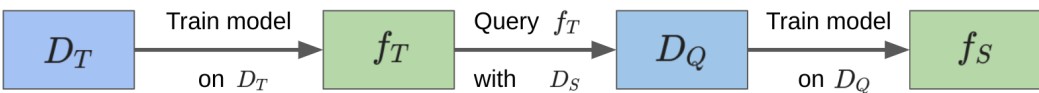

Figure 2: Knowledge distillation is a multi-step training process. A teacher model $f_T$ is first trained on a teacher dataset $D_T$. A dataset of images annotated by the teacher $D_Q$ is generated by querying the teacher $f_T$ on an (unlabeled) student training set $D_S$. Finally, a student model $f_S$ is trained on $D_Q$.

OUT models). To attack a new model $f'$, the adversary computes the probability density function (PDF) of each Gaussian (which we write as $p_{\text{in}}(f'(x)_y)$ and $p_{\text{out}}(f'(x)_y)$), and then computes the likelihood ratio, which serves as the membership score. Our attacks in this paper will be derived from LiRA but adapted to the variety of distillation settings we consider.

## 2.2 Knowledge Distillation

Knowledge distillation [HVD+15; BC14] is a technique for transferring knowledge from a "teacher" model to a "student" model. There are two datasets in knowledge distillation: the teacher dataset $D_T = \{x_i, y_i\}_{i=1}^{n_t}$, with $n_t$ examples, and the student dataset $D_S = \{x_i, y_i\}_{i=1}^{n_s}$, with $n_s$ examples. Distillation begins by training a teacher model $f_T$ on the teacher dataset $D_T$. The teacher model is used to generate (soft) labels for the student dataset, producing a query dataset $D_Q = \{x_i, S(f_T(x_i))\}_{i=1}^{n_s}$, where the softmax function $S$ converts logits into a probability vector. Training on this query dataset then produces a student model $f_S$, the output of distillation. Sometimes, distillation includes "temperature" scaling, where the logits $f_T(x_i)$ are scaled by a temperature parameter $H$, before the softmax is applied. The query dataset will instead be $D_Q = \{x_i, S(f_T(x_i)/H)\}_{i=1}^{n_s}$; setting $H = 1$ recovers the original distillation procedure. We illustrate this process in Figure 2.

**Knowledge distillation in private machine learning.** Prior work has suggested that distillation mitigates prevent privacy attacks. Perhaps the most well known example is the PATE framework [PAEGT16], where distillation is used to reduce an ensemble of teacher models into a single model, in such a way that the final model has provable differential privacy guarantees [DMNS06]. Zheng, Cao, and Wang [ZCW21] and Tang, Mahloujifar, Song, Shejwalkar, Nasr, Houmansadr, and Mittal [TMSSNHM22] construct ensembles of models that are designed to be private and use distillation to condense these models. Mireshghallah, Backurs, Inan, Wutschitz, and Kulkarni [MBIWK22] enforces a differential privacy guarantee by training the teacher and student models with differential privacy. In each of these approaches, distillation is only one component; an underlying ensemble, or provable differential privacy guarantees, may also improve the privacy of the overall approach. In our work, we focus on the distillation procedure itself, and leave to future work the task of designing attacks on these more complicated approaches. Importantly, also, those works which offer provable privacy guarantees will have hard limits on attack effectiveness.

Indeed, other prior work has already suggested that using distillation alone to protect privacy. Shejwalkar and Houmansadr [SH21] propose a defense that relies on a sufficient $\ell_2$ distance between teacher and student examples and limited entropy of the queries. We evaluate this defense in Section 6.4 and Appendix D.1. Mazzone, Heuvel, Huber, Verdecchia, Everts, Hahn, and Peter [MHVEHP22] consider using repeated distillation to prevent membership inference attacks. In our work, we will design stronger membership inference attacks to adaptively evaluate the privacy provided by distillation.

# 3 Threat Model and Experimental Setting

## 3.1 Threat Model

We investigate the ability of distillation to protect against membership inference attacks in three threat models:

1. **Private Teacher.** The teacher dataset $D_T$ is sensitive and the student dataset $D_S$ is nonsensitive. We assume the adversary has knowledge of the student dataset. This threat model

is used in most private machine learning approaches. We evaluate this threat model in the main body of the paper.

2. **Private Student.** The teacher dataset is nonsensitive and the student dataset is sensitive. We assume the adversary has access to the teacher dataset. This threat model may be used to transfer knowledge from a foundation model used as a teacher to a sensitive downstream task [LZZHCZ22]. Because this threat model is less well considered in the privacy literature [SH21], we evaluate this threat model in Appendix D.1.

3. **Self-Distillation.** Self-distillation is commonly used to refer to the setting where the teacher and student datasets are identical. Self-distillation is commonly used when distillation is used to improve model performance or during model compression. We evaluate this threat model in Section D.2.

For each of these threat models, we measure the success of the adversary at performing membership inference on the sensitive dataset, both by measuring the true positive rate (TPR) and false positive rate (FPR) as suggested by Carlini, Chien, Nasr, Song, Terzis, and Tramer [CCNSTT22], and also by investigating the membership inference accuracy on each individual example, as done in Carlini, Jagielski, Zhang, Papernot, Terzis, and Tramer [CJZPTT22]. In general, we will report this per-example membership inference accuracy when presenting the performance of one or two attacks, due to the amount of information this metric conveys, and use ROC curves to compare between more than two.

Beyond distillation's potential uses in privacy, our attacks on distillation have implications for private information leaked during learning-based model extraction attacks [TZJRR16; OSF19; PGSKSG20; JCBKP20], which often resemble model distillation. In model extraction, an adversary uses API access to a target model to reproduce its functionality into the weights of a local model. The target model is analogous to the teacher model in distillation, and the local model is analogous to the student. Then attacks in the Private Teacher threat model can be cast instead as allowing an adversary to inspect their local model to perform membership inference attacks on the target model's training data, without directly querying it. The Private Student threat model has implications for a defensive setting in model extraction, where the target model's owner wants to link queries made to the target model with a model they believe has been extracted from their model.

### 3.2 Experimental Setting

We study four standard datasets for our analysis: CIFAR-10, WikiText103, Purchase100, and Texas100. **On CIFAR-10**, we start with code from the DAWNBench benchmark [CNKZZN-BORZ17] which trains an accurate ResNet-9 model in under 15 seconds; we adapt this code to support model distillation and the subsampling required for LiRA variants. We remove 5275 duplicates from CIFAR-10, using the imagededup library [JLJT19], and split the remaining dataset into a teacher set of 30,000 examples and a student set of 14,725 examples. **On WikiText103**, we used the GPT-2 architecture (with a context window of 256 tokens, 4 heads, 4 layers and an embedding size of 256 dimensions). We split WikiText103 into a teacher set of 500,000 records, and use the remaining records to train the student models. **On Purchase100 and Texas100**, we train single-layer neural networks with hidden layer sizes of 256 and 512, respectively, and subsample the datasets to produce teacher and student sets of 20000 examples each. On Purchase100, student models reach 74-75% accuracy, and on Texas100, they reach 54-55% accuracy. **On all datasets**, we train our models with the cross entropy loss: teacher models are trained with the standard sparse cross entropy loss on the teacher dataset, and student models are trained with a dense cross entropy loss to mimic the soft labels predicted by the teacher. Unless otherwise stated, all LiRA-based attacks use 100 shadow models for calibration, and all figures are produced by running the attack on over 1000 models. Needing to train a large number of shadow models is one limitation of our attacks, and any shadow model-based membership inference attacks; we comment on training efficiency and code for our experiments in Appendix B.

## 4 Evaluating Privacy of the Sensitive Teacher Training Set

The most common way prior work in privacy has used distillation is to improve the privacy of the teacher set. Intuitively, distillation protects the teacher set because the adversary can only interact

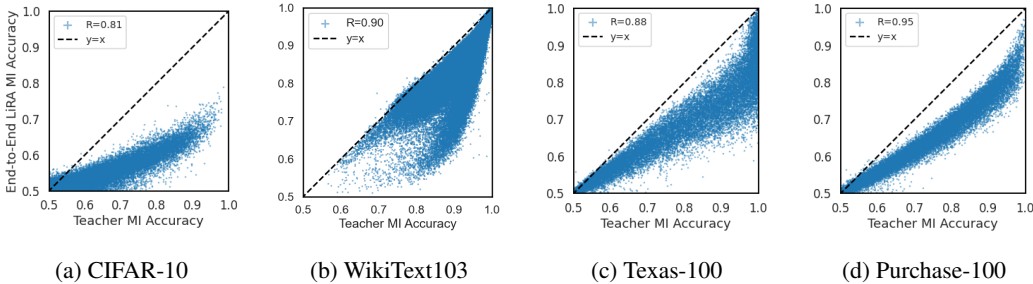

|(a) CIFAR-10|(b) WikiText103|(c) Texas-100|(d) Purchase-100|

Figure 3: **Many data points get no privacy benefits from distillation.** On the x axis, we plot the vulnerability of each teacher example to membership inference before distillation, using teacher models. On the y axis, we plot the vulnerability of each teacher example to attack after distillation, using the End-to-End LiRA strategy. Observe that many data points (very small blue plus signs) lie near the $y = x$ line, which indicates no reduction in vulnerability from distillation.

with the student model, which never sees any teacher examples. In fact, because the student model can be seen as a "post-processing" of the teacher model, the data processing inequality provably implies that attacks can be no more powerful on the student model than they are on the teacher.

Stepping through the distillation process can help us anticipate how distillation can impact the privacy of the teacher set. The first step, training the teacher model, is the most well-understood from a privacy perspective, as the large literature on membership inference applies to the teacher model. In particular, recent work has found that state-of-the-art membership inference attacks are better at attacking some "outlier" examples than other "inlier" examples [CCNSTT22], which we expect to be true in the teacher dataset as well, making these "inlier" examples less vulnerable in later steps, as well.

Subsequent steps of distillation are less well-explored by the privacy literature. The second step of distillation creates the query dataset, which can be seen as "compressing" the teacher model into its responses on these queries. Intuitively, this step is the most important at reducing private information leakage. However, we hypothesize that some queries will capture information about teacher examples, perhaps due to some similarity between the queries and teacher examples. In the final step, the student model is trained on the query set. While this step cannot contain more sensitive information than the queries themselves, it is possible that this step makes that information easier to *discover*, perhaps by interpolating between the queries.

## 4.1 Performing Membership Inference on Teacher Examples

We now show that membership inference attacks work surprisingly well on distilled student models.

We first propose the "Transfer LiRA" attack, a simple extension of LiRA to the distillation setting. In this attack, we train shadow *teacher* models. We calibrate the IN and OUT Gaussians for LiRA on these shadow teacher models, and then run the attack by applying it directly to the student models. Notice that in this attack, distillation never happens in training shadow models—the success of this attack relies on the similarity between teacher and student models' predictions.

To capture the information loss because Transfer LiRA does not use distillation in any way, we also propose a second attack, which we call "End-to-End LiRA", where we instead train shadow models with the entire distillation procedure, first training shadow teacher models, and then distilling these teacher models into shadow *student* models. We then calibrate LiRA using these shadow student models. Note that performing this attack requires knowledge of the student training set to query the shadow teacher models and train the shadow student models. This is in contrast to Transfer LiRA which does not, although it still requires access to in-distribution data to train teacher models.

**Distillation provides limited privacy.** In Figure 3, we plot the change in per-example attack success rate on distilled models. We compare each example's vulnerability to LiRA on the teacher model (i.e. vulnerability without distillation) on the x-axis, to each example's vulnerability to End-to-End LiRA (i.e. vulnerability after distillation) on the y-axis. In the Appendix, we provide plots for

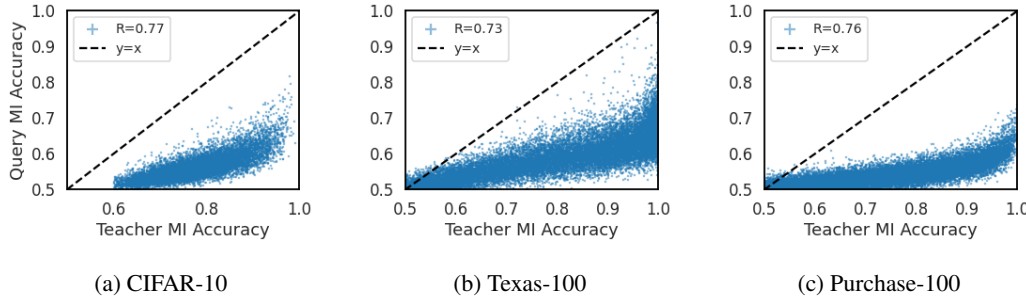

| (a) CIFAR-10 | (b) Texas-100 | (c) Purchase-100 |

Figure 4: Per-example membership inference accuracy using only student queries, compared to LiRA accuracy on the teacher model. Membership inference accuracies are non-trivial, and remain high for some examples. Effectiveness is dataset- and example-dependent, with Texas-100 having the most vulnerable examples in the teacher model remaining vulnerable to attacks based on teacher predictions.

Transfer LiRA (Figure 7), and ROC curves for each strategy (Figure 8). We also find our attack significantly outperforms a simple logit threshold baseline, similar to the weaker attacks used in prior work to evaluate distillation (Figure 12).

For a large fraction of teacher examples on each dataset, we find End-to-End LiRA achieves nearly the same membership inference performance as directly attacking the teacher model. In other words, many teacher examples do not observe *any* privacy benefits from distillation, despite student models never directly seeing these teacher examples! While the average membership inference accuracy (and TPR at low FPR) do decrease, 5% of examples' vulnerability drop by less than 8 percentage points on CIFAR-10, 5 points on Purchase-100, and 4 points on Texas-100, and these examples have a variety of teacher vulnerabilities. Because privacy is a worst-case guarantee, distillation provides limited privacy benefits.

We note two other interesting takeaways from these results. First, the per-example student attack success rate has a high variance. This variance is partly due to statistical uncertainty (although each example's attack success rate is computed with over 1000 models, giving each coordinate a standard deviation of 1.5 percentage points for both the $x$ and $y$ axes), but more interestingly, some examples do see significant privacy benefits from distillation, even controlling for the original model's vulnerability. We investigate duplication as one potential cause for this variance in Section 6.1. Second, each plot has a positive correlation, meaning that examples that are more vulnerable to attack on the teacher model attack also tend to be more vulnerable after distillation. As a result, reducing the vulnerability of the teacher model, perhaps using techniques such as differential privacy, are likely to improve the student's privacy.

## 5 Tracing Teacher Privacy Leakage Through Student Queries

It is surprising that membership inference still performs well on distilled student models, despite these models never directly using any of the teacher data. We now investigate *why*. For a membership inference attack to be successful on the student model, it must be the case that the student queries reveal some membership information about the teacher examples. However, to rigorously evaluate how this information is encoded in the student queries, we will turn our attention to designing a new attack that only has access to the student query dataset $D_Q$ (and, in particular, has no knowledge of the teacher model's predictions on the teacher examples). To the best of our knowledge, this attack is also *the only membership inference attack in the literature* which does not require querying a model directly on an example (or on algorithmically derived examples) to predict its membership status.

**An indirect attack using student queries.** We adapt LiRA to a setting where only the student query scores are available to the adversary. Because of this limitation, the adversary can only rely on information about teacher examples that is indirectly contained in the student query scores. To adapt the attack to this setting, we use the same approach that LiRA uses to combine queries on

multiple "augmentations" of an example: multiplying their likelihood ratios to produce an aggregate membership score. Concretely, for each teacher example $z_j^T = (x_j^T, y_j^T)$, we fit a Gaussian distribution to the logits of each student example $z_i^S = (x_i^S, y_i^S)$, when $z_j^T$ is either IN or OUT. We write the PDFs of these distributions as $p_{j,i}^{IN}$ and $p_{j,i}^{OUT}$, so that the joint PDF of the IN Gaussians is $p_j^{IN} = \Pi_i p_{j,i}^{IN}$ and similarly for the OUT Gaussians. This natural adjustment allows us to infer membership of the teacher examples using the student examples.

However, we find that this direct adaptation of LiRA tends to have poor performance, so we propose two modifications which significantly improve the attack. First, we find that a teacher example tends to have more influence on the logit corresponding to the teacher label $y_j^T$ than the student label $y_i^S$, and so we choose to calibrate LiRA using the teacher label logit rather than the student label logit. Second, there tend to be many uninformative student examples for each teacher example, so we filter the student queries for all teacher examples. We filter these student queries by selecting only those with the largest mean gap $|\mu_{j,i}^{IN} - \mu_{j,i}^{OUT}|$, which are most informative about the teacher. This removes most of the noise from our LiRA adaptation, improving it especially when we train few shadow models.

**Student queries leak private information.** In Figure 4, we plot the per-example membership inference accuracy after distillation, using our student query attack with 4000 shadow models. We consider all datasets except WikiText103, which we omit due to the high computational cost of our attack. Our results are similar to when we attacked the student models: MIA vulnerability reduces on average, but many examples maintain nearly identical membership inference attack success rate using only the predictions on these indirect student queries. Cases where this attack achieves high membership inference accuracy indicate that a large amount of information about the teacher set is encoded in the student queries.

Observations we noticed for our student model attacks persist for this query-based attack. Examples which are more vulnerable to attack in the teacher model also tend to be more vulnerable using teacher model predictions. This is true on all datasets, although we also find the decay in vulnerability with queries to be dataset-dependent and example-dependent. Within each dataset, there is a high variance in student model membership inference accuracy.

We also remark on the counterintuitive fact that our prior attack with access to the student models outperformed this attack, despite the fact that the student model is just a post-processing of these student queries. This implies that the process of training a student model makes it easier to extract private information from these student queries, by interpolating between the student queries. In the following subsections, we will investigate some factors that can impact the success of the student query attack, and what this tells us about distillation's dark knowledge.

**Ablation.** In Figure 5a, we ablate the modifications we made for our student query attack. We show both LiRA scores (logits for the student label) and Label scores (logits for the teacher label). We also show attacks with "All", without filtering, and "Filtered" scores, with filtering to 10 student examples. Our modifications significantly improve membership inference: without either modification, the attack is no better than random chance. Applying both modifications achieves a TPR of $10^{-3}$ at a FPR of $10^{-4}$.

**Qualitative analysis.** Our ablations also shed light on where membership information is contained in distillation's dark knowledge: predominantly in the logit corresponding to the teacher label within only a few student examples. This corroborates our intuition from the parrot in Figure 1. There, only two student queries were also in the bird class, emphasizing the importance of label scores. We inspect other vulnerable examples in Figure 9 in Appendix C, finding some teacher examples whose most informative students mostly belong to their same class.

# 6 Factors Influencing Teacher Privacy Leakage

So far, we have designed two types of attack: the attack from Section 4 which uses the student model, and the attack from Section 5 which uses the student queries. These attacks find a surprising amount of leakage which could not have been identified by prior attacks. Now, we investigate more deeply the factors which may influence the leakage of the teacher.

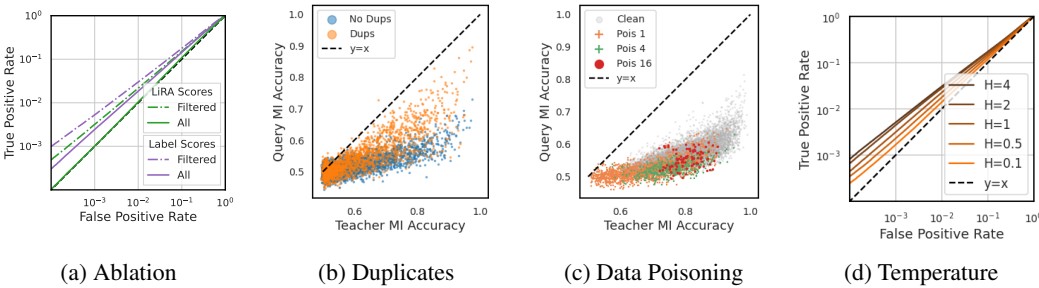

| (a) Ablation | (b) Duplicates | (c) Data Poisoning | (d) Temperature |

Figure 5: Additional investigations into the success of our attacks reveal: a) label scoring and score filtering are important for improving attack success; b) duplication between the teacher and student datasets increases privacy risk; c) data poisoning attacks amplify the performance of our indirect attack; d) temperature scaling causes mild changes in privacy vulnerability. All results are on CIFAR-10.

## 6.1 Student-Teacher Duplication

Unlike the preceding indirect queries, where student examples only share high-level similarities with their teacher examples, we now investigate how increased student-teacher similarity impacts the performance of our student query attack. To do so, we reintroduce the near-duplicates we removed from CIFAR-10, and compare the attack success rate between the duplicated and deduplicated student query attacks, on those teacher examples with duplicate student examples. Intuitively, student examples which are duplicates of teacher examples will carry membership information in their queries, which is what we find in Figure 5b ($p < 10^{-15}$ using a Chow Test). This indicates that deduplication significantly reduces privacy risk, although indirect queries can still carry membership information. In the Appendix, we show student attacks are also worsened by duplication in Figure 11.

## 6.2 Teacher Set Poisoning

Tramèr, Shokri, San Joaquin, Le, Jagielski, Hong, and Carlini [TSSJLJHC22] and Chen, Shen, Shen, Wang, and Zhang [CSSWZ22] find that data poisoning attacks added into a training set can amplify membership inference vulnerability for other examples in the training set. Here, we investigate how distillation interacts with this effect; i.e., we measure whether poisoning in the teacher set can increase an example's vulnerability in the teacher predictions. In Figure 5c, we evaluate this using our student query attack on CIFAR-10 using the label flipping poisoning strategy from prior work.

With higher poisoning counts, many examples have higher teacher membership inference accuracy, i.e., they shift to the right on the $x$-axis; this is exactly in line with prior work. However, interestingly, we see that poisoning does not impact the relationship between teacher vulnerability and student vulnerability—they move upward on the $y$-axis identically to examples which were not poisoned.

We remark that our poisoning attack does not specifically target the student query set; it is an interesting open question whether poisoning attacks exist that could increase membership leakage on the student model, but with less (or more!) impact on the teacher model's vulnerability. Section 6.1 hints at such a strategy, if the adversary can poison the student set instead: if an adversary adds duplicates of target examples to the student set, the resulting student queries will increase risk on those target teacher examples.

## 6.3 Temperature Scaling

When introducing knowledge distillation, Hinton, Vinyals, Dean, et al. [HVD+15] proposed a modification known as temperature scaling. Temperature scaling makes two simple changes to distillation: (1) introducing a temperature hyperparameter $H$, which when increased, rescales the logits and flattens the resulting probability distribution of student queries, and (2) rescaling gradients by a $1/H^2$ factor. Though normally set to 1, modifying this parameter can improve performance. It is natural to consider whether such manipulation of the teacher outputs might improve empirical

privacy; indeed, Shejwalkar and Houmansadr [SH21] evaluate their distillation-based defense with various temperatures, and find that high temperature reduces private information.

We use our strong End-to-End LiRA to evaluate the empirical privacy of student models trained with $H \in [0.1, 4]$, shown in Figure 5d. We find lower temperatures are mildly less vulnerable than higher temperatures, reducing TPR by a factor of roughly 4 at an FPR of $10^{-3}$ when $H$ decreases from 4 to 0.1; model accuracy at all of these temperatures is similar to the baseline of $H = 1$. While this trend is small, it is also intuitive: in Section 5, we found that all logit values carry important membership information. When $H$ is small, the entropy of query probabilities decreases, reducing information captured by these probabilities. We hypothesize that the reverse effect observed by Shejwalkar and Houmansadr [SH21] may be a result of the accuracy decrease they found resulted from high temperatures (i.e. up to $H = 10$), which we do not observe.

Finally, we remark on an interesting gap between an adversary with access to student queries and one who only has access to a student model. With access to temperature-scaled student logits, an adversary can multiply by $H$ to reverse the scaling—temperature scaling cannot make attacks harder in this threat model.

However, it does appear that low temperatures lead to weaker attacks on the student model, which we do not know how to reverse. Indeed, a natural way to adaptively attack a temperature-scaled student model is to rescale the student model's logits, but our LiRA accounts for this, as LiRA is scale-invariant. Adaptively attacking temperature-scaled models is an interesting open question.

## 6.4 Student Distribution Drift

In cases where the teacher dataset is sensitive, nonsensitive student data from the same distribution may be difficult to find, leading to distribution drift between the teacher and student datasets. We now investigate whether this impacts attack success. From our results on student-teacher duplication, we may expect the signal to diminish as distributions become more different. To evaluate this, we consider a CIFAR-10 teacher model distilled with a CIFAR-100 student set; CIFAR-10 and CIFAR-100 have no class overlap but have the same size. Interestingly, we find that distilling with this different dataset actually *improves* attack performance, despite the resulting model being somewhat less accurate than after distilling on in-distribution data. We find that this is due to a higher entropy of the teacher model predictions, which can encode more membership information about the teacher examples. We run an experiment in which we train on the lowest entropy and highest entropy predictions from CIFAR-100, as well as random student queries. All experiments

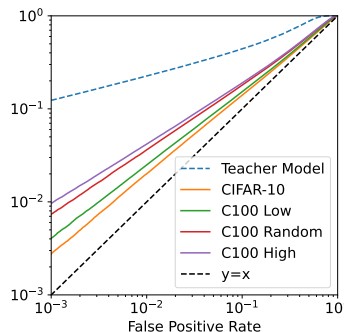

Figure 6: A CIFAR-100 student set results in more privacy leakage than a CIFAR-10 student set, a result of the increased entropy of the CIFAR-100 predictions.

use the same student set sizes. We present the results in Figure 6. The CIFAR-10 student still results in the lowest leakage, and increasing entropy of the CIFAR-100 examples also increases the leakage monotonically. Note that the CIFAR-10 student predictions have a mean entropy of 0.24, while the lowest leakage CIFAR-100 examples have a mean entropy of 0.39. Our results here can also be taken as an attack on the full Shejwalkar and Houmansadr [SH21] defense, which removes high entropy and duplicate student examples. Our strong attacks allow us to better evaluate their defense.

## 6.5 Accuracy of the Student Model

It is common for privacy preserving training algorithms to exhibit a "privacy-utility" tradeoff: the stronger the privacy enforced by the model, the worse the model's predictive performance. By piecing together the accuracies of models trained throughout Section 6, we find that differences in attack performance cannot be explained by changes in model accuracy. For example, we find in Section 6.3 that increasing temperature increases vulnerability to our attacks but does not significantly impact model accuracy (maintaining roughly 88% accuracy for the student). Our results with distribution drift in Section 6.4 show that distribution shift leads to higher vulnerability to our attacks,

but reduces model accuracy (from 88% to roughly 85% student model accuracy with a CIFAR-100 distilled student). Finally, we find in Section 5b that deduplication reduces vulnerability to our attacks, but actually increases model accuracy slightly (from roughly 87.5% to 88% student accuracy)! In summary, depending on the source of a model performance change, increased vulnerability can coincide with increased, decreased, or unchanged model performance!

# 7 Conclusion

In this work, we have used membership inference attacks to empirically evaluate to what extent knowledge distillation protects the privacy of training data. For those interested in using model distillation to improve privacy, our work offers three main design considerations. First, deduplicating the teacher set with respect to the student set is necessary to reduce risk on duplicated examples. Second, while average-case privacy can significantly improve after distillation, the worst-case vulnerable examples often see only marginal benefits. As a result, techniques which make the teacher model more private, such as differentially private training, should be seen as complementary to distillation. And third, the student training set should not be seen to have improved privacy as a result of model distillation. Our work also offers insights into the privacy properties of distillation's dark knowledge, which may be of broader interest. Our results also imply privacy attacks on model extraction attacks [TZJRR16; JCBKP20] which rely on similar algorithms to knowledge distillation.

**Limitations**

The main limitation of our work is the strong threat model under which our attacks work. We use the same threat model as the "online" version of the LiRA [CCNSTT22] attack. This attack assumes access to the target examples before training a model, and, for the End-to-End LiRA attack, access to the student dataset. We use this strong threat model to assess worst-case vulnerability, as prior work has evaluated distillation under weaker attacks. Another limitation of the attacks is the large running time of the student query attack from Section 5, which requires thousands of shadow models to obtain good performance. While generally impractical, we prefer to position this attack as a way of explaining why distillation propagates membership information, and leave future work to attempt to improve the attack's efficiency.

**Acknowledgements**

We would like to thank Andreas Terzis and Andreas Schou for helpful comments on our work.

# References

| | |
|---|---|
| [BC14] | J. Ba and R. Caruana. "Do deep nets really need to be deep?" In: *Advances in neural information processing systems* 27 (2014) (cit. on pp. 1, 3). |
| [CCNSTT22] | N. Carlini, S. Chien, M. Nasr, S. Song, A. Terzis, and F. Tramer. "Membership inference attacks from first principles". In: *2022 IEEE Symposium on Security and Privacy (SP)*. IEEE. 2022, pp. 1897–1914 (cit. on pp. 1, 2, 4, 5, 10). |
| [CCTCP21] | C. A. Choquette-Choo, F. Tramer, N. Carlini, and N. Papernot. "Label-only membership inference attacks". In: *International conference on machine learning*. PMLR. 2021, pp. 1964–1974 (cit. on p. 2). |
| [CJZPTT22] | N. Carlini, M. Jagielski, C. Zhang, N. Papernot, A. Terzis, and F. Tramer. "The Privacy Onion Effect: Memorization is Relative". In: *Advances in Neural Information Processing Systems*. Ed. by A. H. Oh, A. Agarwal, D. Belgrave, and K. Cho. 2022. URL: https://openreview.net/forum?id=ErUlLrGaVEU (cit. on p. 4). |
| [CNKZZNBORZ17] | C. Coleman, D. Narayanan, D. Kang, T. Zhao, J. Zhang, L. Nardi, P. Bailis, K. Olukotun, C. Ré, and M. Zaharia. "Dawnbench: An end-to-end deep learning benchmark and competition". In: *Training* 100.101 (2017), p. 102 (cit. on p. 4). |

[CSSWZ22]        Y. Chen, C. Shen, Y. Shen, C. Wang, and Y. Zhang. "Amplifying Membership Exposure via Data Poisoning". In: *Advances in Neural Information Processing Systems*. Ed. by A. H. Oh, A. Agarwal, D. Belgrave, and K. Cho. 2022 (cit. on pp. 2, 8).

[CTWJHVLRBSE+21]  N. Carlini, F. Tramer, E. Wallace, M. Jagielski, A. Herbert-Voss, K. Lee, A. Roberts, T. Brown, D. Song, U. Erlingsson, et al. "Extracting training data from large language models". In: *30th USENIX Security Symposium (USENIX Security 21)*. 2021, pp. 2633–2650 (cit. on p. 2).

[DMNS06]         C. Dwork, F. McSherry, K. Nissim, and A. Smith. "Calibrating noise to sensitivity in private data analysis". In: *Theory of cryptography conference*. Springer. 2006, pp. 265–284 (cit. on pp. 1–3).

[FJR15]          M. Fredrikson, S. Jha, and T. Ristenpart. "Model inversion attacks that exploit confidence information and basic countermeasures". In: *Proceedings of the 22nd ACM SIGSAC conference on computer and communications security*. 2015, pp. 1322–1333 (cit. on p. 2).

[FLTIA18]        T. Furlanello, Z. Lipton, M. Tschannen, L. Itti, and A. Anandkumar. "Born again neural networks". In: *International Conference on Machine Learning*. PMLR. 2018, pp. 1607–1616 (cit. on p. 14).

[GWYGB18]        K. Ganju, Q. Wang, W. Yang, C. A. Gunter, and N. Borisov. "Property inference attacks on fully connected neural networks using permutation invariant representations". In: *Proceedings of the 2018 ACM SIGSAC conference on computer and communications security*. 2018, pp. 619–633 (cit. on p. 2).

[HVD+15]         G. Hinton, O. Vinyals, J. Dean, et al. "Distilling the knowledge in a neural network". In: *arXiv preprint arXiv:1503.02531* 2.7 (2015) (cit. on pp. 1, 3, 8).

[JCBKP20]        M. Jagielski, N. Carlini, D. Berthelot, A. Kurakin, and N. Papernot. "High accuracy and high fidelity extraction of neural networks". In: *29th USENIX security symposium (USENIX Security 20)*. 2020, pp. 1345–1362 (cit. on pp. 4, 10).

[JLJT19]         T. Jain, C. Lennan, Z. John, and D. Tran. *Imagededup*. `https://github.com/idealo/imagededup`. 2019 (cit. on p. 4).

[JWKGE20]        B. Jayaraman, L. Wang, K. Knipmeyer, Q. Gu, and D. Evans. "Revisiting membership inference under realistic assumptions". In: *arXiv preprint arXiv:2005.10881* (2020) (cit. on p. 2).

[KPK18]          J. Kim, S. Park, and N. Kwak. "Paraphrasing complex network: Network compression via factor transfer". In: *Advances in neural information processing systems* 31 (2018) (cit. on p. 1).

[LBWBWTGC18]     Y. Long, V. Bindschaedler, L. Wang, D. Bu, X. Wang, H. Tang, C. A. Gunter, and K. Chen. "Understanding membership inferences on well-generalized learning models". In: *arXiv preprint arXiv:1802.04889* (2018) (cit. on p. 2).

[LZ21]           Z. Li and Y. Zhang. "Membership leakage in label-only exposures". In: *Proceedings of the 2021 ACM SIGSAC Conference on Computer and Communications Security*. 2021, pp. 880–895 (cit. on p. 2).

[LZZHCZ22]       C. Liang, S. Zuo, Q. Zhang, P. He, W. Chen, and T. Zhao. "Less is More: Task-aware Layer-wise Distillation for Language Model Compression". In: *arXiv preprint arXiv:2210.01351* (2022) (cit. on p. 4).

[MBIWK22]        F. Mireshghallah, A. Backurs, H. A. Inan, L. Wutschitz, and J. Kulkarni. "Differentially Private Model Compression". In: *arXiv preprint arXiv:2206.01838* (2022) (cit. on p. 3).

[MHHVEHP22]      F. Mazzone, L. van den Heuvel, M. Huber, C. Verdecchia, M. Everts, F. Hahn, and A. Peter. "Repeated Knowledge Distillation with Confidence Masking to Mitigate Membership Inference Attacks". In: *Proceedings of the 15th ACM Workshop on Artificial Intelligence and Security*. AISec'22. Los Angeles, CA, USA: Association for Computing Machinery, 2022,

13–24. ISBN: 9781450398800. URL: https://doi.org/10.1145/3560830.3563721 (cit. on pp. 1, 3).

[OSF19]        T. Orekondy, B. Schiele, and M. Fritz. "Knockoff nets: Stealing functionality of black-box models". In: *Proceedings of the IEEE/CVF conference on computer vision and pattern recognition*. 2019, pp. 4954–4963 (cit. on p. 4).

[PAEGT16]      N. Papernot, M. Abadi, U. Erlingsson, I. Goodfellow, and K. Talwar. "Semi-supervised knowledge transfer for deep learning from private training data". In: *arXiv preprint arXiv:1610.05755* (2016) (cit. on pp. 1, 3).

[PGSKSG20]     S. Pal, Y. Gupta, A. Shukla, A. Kanade, S. Shevade, and V. Ganapathy. "Activethief: Model extraction using active learning and unannotated public data". In: *Proceedings of the AAAI Conference on Artificial Intelligence*. Vol. 34. 2020, pp. 865–872 (cit. on p. 4).

[PPA18]        A. Polino, R. Pascanu, and D. Alistarh. "Model compression via distillation and quantization". In: *arXiv preprint arXiv:1802.05668* (2018) (cit. on p. 1).

[SCGL19]       S. Sun, Y. Cheng, Z. Gan, and J. Liu. "Patient knowledge distillation for bert model compression". In: *arXiv preprint arXiv:1908.09355* (2019) (cit. on p. 1).

[SDSOJ19]      A. Sablayrolles, M. Douze, C. Schmid, Y. Ollivier, and H. Jégou. "Whitebox vs black-box: Bayes optimal strategies for membership inference". In: *International Conference on Machine Learning*. PMLR. 2019, pp. 5558–5567 (cit. on p. 2).

[SH21]         V. Shejwalkar and A. Houmansadr. "Membership privacy for machine learning models through knowledge transfer". In: *Proceedings of the AAAI Conference on Artificial Intelligence*. Vol. 35. 2021, pp. 9549–9557 (cit. on pp. 1, 3, 4, 9, 13).

[SSSS17]       R. Shokri, M. Stronati, C. Song, and V. Shmatikov. "Membership inference attacks against machine learning models". In: *2017 IEEE symposium on security and privacy (SP)*. IEEE. 2017, pp. 3–18 (cit. on pp. 1, 2).

[TMSSNHM22]    X. Tang, S. Mahloujifar, L. Song, V. Shejwalkar, M. Nasr, A. Houmansadr, and P. Mittal. "Mitigating membership inference attacks by {Self-Distillation} through a novel ensemble architecture". In: *31st USENIX Security Symposium (USENIX Security 22)*. 2022, pp. 1433–1450 (cit. on pp. 1, 3).

[TSSJLJHC22]   F. Tramèr, R. Shokri, A. San Joaquin, H. Le, M. Jagielski, S. Hong, and N. Carlini. "Truth Serum: Poisoning Machine Learning Models to Reveal Their Secrets". In: *Proceedings of the 2022 ACM SIGSAC Conference on Computer and Communications Security*. CCS '22. Los Angeles, CA, USA: Association for Computing Machinery, 2022, 2779–2792. ISBN: 9781450394505. URL: https://doi.org/10.1145/3548606.3560554 (cit. on p. 8).

[TZJRR16]      F. Tramèr, F. Zhang, A. Juels, M. K. Reiter, and T. Ristenpart. "Stealing machine learning models via prediction {APIs}". In: *25th USENIX security symposium (USENIX Security 16)*. 2016, pp. 601–618 (cit. on pp. 2, 4, 10).

[WBKBGGG22]    Y. Wen, A. Bansal, H. Kazemi, E. Borgnia, M. Goldblum, J. Geiping, and T. Goldstein. "Canary in a Coalmine: Better Membership Inference with Ensembled Adversarial Queries". In: *arXiv preprint arXiv:2210.10750* (2022) (cit. on p. 2).

[WGCS21]       L. Watson, C. Guo, G. Cormode, and A. Sablayrolles. "On the importance of difficulty calibration in membership inference attacks". In: *arXiv preprint arXiv:2111.08440* (2021) (cit. on p. 2).

[XLHL20]       Q. Xie, M.-T. Luong, E. Hovy, and Q. V. Le. "Self-training with noisy student improves imagenet classification". In: *Proceedings of the IEEE/CVF conference on computer vision and pattern recognition*. 2020, pp. 10687–10698 (cit. on pp. 1, 14).

[YGFJ18]     S. Yeom, I. Giacomelli, M. Fredrikson, and S. Jha. "Privacy risk in machine learning: Analyzing the connection to overfitting". In: *2018 IEEE 31st computer security foundations symposium (CSF)*. IEEE. 2018, pp. 268–282 (cit. on pp. 1, 2, 13).

[ZCW21]      J. Zheng, Y. Cao, and H. Wang. "Resisting membership inference attacks through knowledge distillation". In: *Neurocomputing* 452 (2021), pp. 114–126. ISSN: 0925-2312. URL: https://www.sciencedirect.com/science/article/pii/S0925231221006329 (cit. on p. 3).

[ZK16]       S. Zagoruyko and N. Komodakis. "Paying more attention to attention: Improving the performance of convolutional neural networks via attention transfer". In: *arXiv preprint arXiv:1612.03928* (2016) (cit. on p. 1).

## A  Broader Impact

Our work designs privacy attacks, which have the potential to cause harm. However, by making the vulnerabilities in existing approaches known, and more rigorously evaluating the risk to users, our work is a necessary step to designing stronger mitigations in the future.

## B  More Experiment Details

All of our results on CIFAR-10 make use of fewer than 30000 trained models. While a very large number of models, the fast, publicly available training code we use allows us to train this number of models in fewer than 1 GPU-week (although we decrease the wall-clock time by parallelizing over 4 GPUs). Our results on Purchase-100 and Texas-100 also use simple models, taking under 1 minute to train (we train all models for 20 epochs with SGD with a learning rate of 0.01 and momentum parameter of 0.99, which we found to maximize performance over our hyperparameter sweep). We train 8000 of these models for our analysis, taking fewer than 1 GPU-week for each of these datasets. Our most expensive attack, relying on only student queries, starts to outperform random guessing with as few as 100 models, which can be trained on 1 GPU in two hours on all three of these datasets. Unfortunately, we are unable to make our code public at this time due to organizational constraints.

## C  Extended Results on Teacher Dataset Privacy

We plot the effectiveness of Transfer LiRA in Figure 7. ROC curves for our student attacks are found in Figure 8. Further qualitative examples can be found in Figure 9. Ablation of score information with and without duplicates is plotted in Figure 10. Per-example student attack success rates for CIFAR-10 with duplicates are found in Figure 11. In Figure 12, we compare our student model attacks against a simple logit threshold baseline, similar to the loss thresholding attack designed by Yeom, Giacomelli, Fredrikson, and Jha [YGFJ18], which was used to evaluate distillation privacy in Shejwalkar and Houmansadr [SH21].

## D  Privacy of Student Training Set

Having evaluated the Private Teacher threat model, we now turn to the Private Student and Self-Distillation threat models, which we will consider simultaneously. The Private Student threat model can be used to perform knowledge transfer from large, general purpose models to task-specific models, by querying on (sensitive) task-specific student data. Self-distillation is often used in applications of distillation to compress models and improve their performance.

### D.1  Private Student

The private student threat model does not involve data minimization, unlike the private teacher threat model; the empirical privacy we investigate here comes instead from an adversary having limited knowledge of the specifics of the teacher model. That is, the question we investigate is: how much

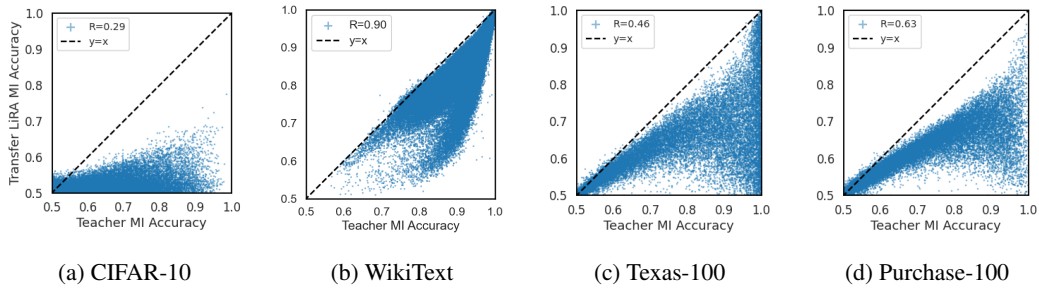

(a) CIFAR-10         (b) WikiText         (c) Texas-100         (d) Purchase-100

Figure 7: **Many data points do not get privacy benefits from distillation.** With the x axis, we plot the vulnerability of each teacher example to attack before distillation, using teacher models. With the y axis, we plot the vulnerability to attack after distillation, using the Transfer LiRA strategy to attack student models. Observe that many data points lie near the $y = x$ line, which indicates no reduction in vulnerability from distillation.

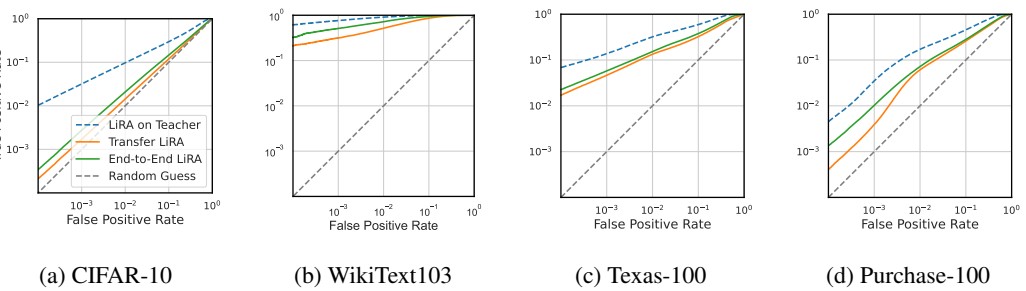

(a) CIFAR-10         (b) WikiText103         (c) Texas-100         (d) Purchase-100

Figure 8: ROC curves for our attacks on student models.

does the adversary need to know about the teacher model to get reliable attacks on the private student dataset?

We consider three levels of adversarial knowledge: **Known Teacher**, where the adversary knows the precise teacher model used to query the student examples; **Unknown Teacher**, where the adversary knows the teacher model is one of a small subset of models; and **Surrogate**, where the adversary can only collect similar data, to train their own surrogate teacher models. Both the Known and Unknown Teacher settings reflect a world where the teacher model is one of a small number of general purpose public models, such as a large language model. The Surrogate setting requires the adversary to train their own copy.

We run the LiRA variants in a number of these settings on the CIFAR-10 dataset, calibrated to the knowledge the adversary has (for example, in the Surrogate threat model, the adversary trains their own teacher models, and trains a number of shadow student models to calibrate LiRA). We plot our results in Figure 13a, and find that, as expected, less knowledge about the teacher model reduces the adversary's success at membership inference. However, even the weakest threat model, Surrogate, allows for powerful attacks, with a TPR as large as $10^{-2}$ at a FPR of $10^{-3}$.

### D.2 Privacy of Self-Distillation

Having considered the privacy of the student and teacher datasets independently, we now investigate the common self-distillation setting [FLTIA18; XLHL20], where the student and teachers are identical. Given that duplicate examples in the student set carry membership information of teacher examples (Section 6.1), and student examples themselves are not well protected by distillation (Section D.1), we do not expect self-distillation to reduce privacy risk significantly. However, a common technique in self-distillation is to train the student on a loss function which combines the cross entropy loss on the query dataset $\ell_Q$ with the cross entropy loss on the student examples' original "hard labels" $\ell_S$. We write $\ell_\alpha = \alpha \ell_Q + (1 - \alpha)\ell_S$, so that $\alpha = 1$ recovers the standard distillation

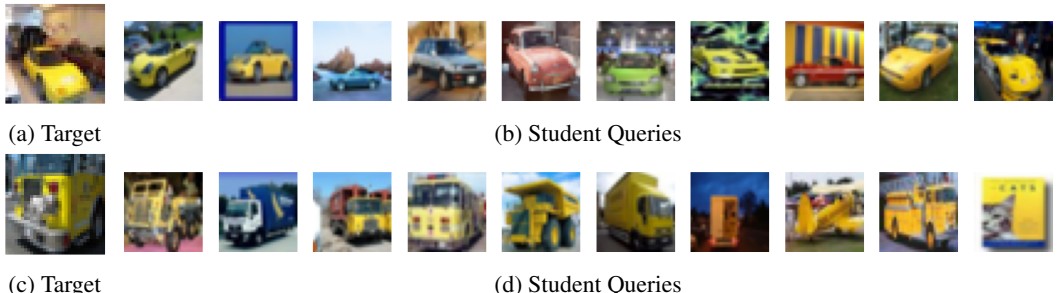

(a) Target                                    (b) Student Queries

(c) Target                                    (d) Student Queries

Figure 9: Two examples of target examples for which the most informative student queries are predominantly in the same class. The only exception is the eighth student query in (d) for the yellow truck in (c), which is an airplane. The filtered attack using the displayed student queries reaches 78% accuracy on the yellow automobile in (a), and 74% accuracy on the yellow truck in (c).

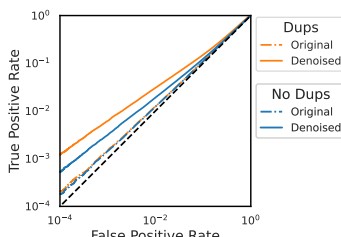

Figure 10: The impact of denoising on duplicated and deduplicated teacher attacks.

objective, while $\alpha = 0$ recovers the standard cross entropy loss (as if there was never a teacher model).

To evaluate self-distillation, we run LiRA by training shadow student models with the entire self-distillation algorithm, using identical datasets for each pair of teacher and student shadow models. We perform calibration on these shadow student models, and plot our results at a range of $\alpha$ values in Figure 13b. While we don't observe a large effect, it appears that larger $\alpha$ (that is, heavier reliance on the distillation loss function) results in better attacks. This is likely because relying on the distillation loss function reinforces the memorization from the teacher even further in the second round of training on the student.

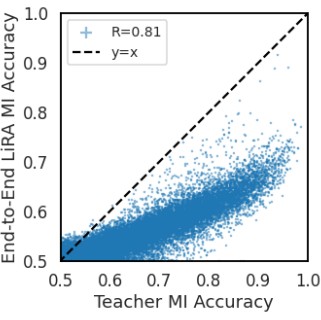

Figure 11: Duplication also has an impact on CIFAR-10 student attacks. Compare with Figure 3a.

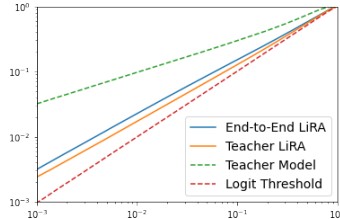

Figure 12: Our attacks outperform a simple logit threshold baseline attack, used by prior work.

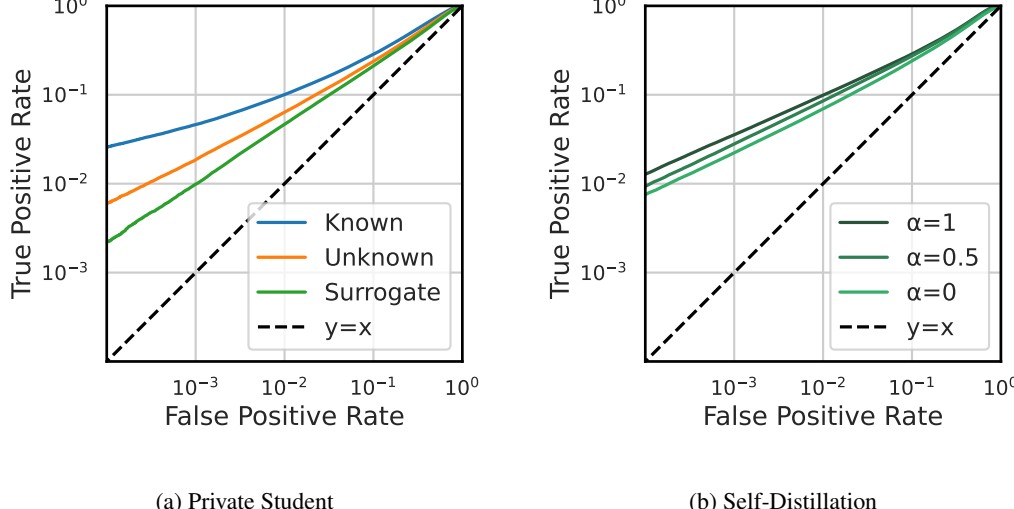

(a) Private Student

(b) Self-Distillation

Figure 13: *Distillation has limited ability to prevent membership inference* either a) on sensitive student examples, or b) in self-distillation. However, reducing the knowledge available to the adversary seems to help in the Private Student threat model. Results for both on CIFAR-10.

