# OpenReview forum: "Students Parrot Their Teachers: Membership Inference on Model Distillation"
_NeurIPS.cc/2023/Conference — NeurIPS 2023 oral_

### Official Review · Reviewer_xTVE · 2023-07-04

**Soundness:** 4 excellent
**Presentation:** 3 good
**Contribution:** 3 good
**Rating:** 8
**Confidence:** 4

**Summary:**

Multiple previous works have proposed knowledge distillation techniques to distill the knowledge of a teacher trained on sensitive data into a student model which is supposedly protected against membership inference attacks. This paper proposes a new membership inference attack to perform membership inference attacks on the student model. Attacking the student models, it is shown that knowledge distillation provides only limited privacy against membership inference attacks. Having access to the dataset used to train the student model but not the student model itself, it is shown that this suffices to attack the private data of the teacher model.

**Strengths:**

- **Originality:** The paper is novel and proposes a novel approach for membership inference attacks on student models.
- **Quality:** The experimental results seem sound and the claims of the paper are well supported.
- **Clarity:** Overall, the paper is mostly well written and well organized.
- **Significance:** With previous work having proposed knowledge distillation as a defense technique against membership inference, the results of the paper are important and most likely of high impact to other researchers.

**Weaknesses:**

**Clarity:**
- The setting and the assumptions made for the proposed attacks are not very clear, and it would be helpful for the reader to explain the assumptions and information available to the attacker in more detail.
- unfortunately, the code used to run the experiments will not be made public, which hinders reproducibility

Misc:
- Line 94: "mitigates prevent". There seems to be some missing or additional word.
- maybe you could mention in the captions that “+” in the plots means the positive correlation. Otherwise, it is a bit confusing, if the reader is searching for a blue plus in the plots.

**Questions:**

**Q1:** The per example membership inference attack success rate is calculated over 1000 models. However, how do these models differ? Were they trained on the same data but with different seeds? If so, aren't these models behaving very similar?
**Q2:** In line 241 the indirect attack is described. However, the procedure of the attack is not quite clear to me. As far as I understood, the attacker has no knowledge about the teacher model's predictions on the teacher examples (line 231). Could you clarify what the assumptions made for this attack are and on how the attacker is fitting the Gaussians in this case?

**Limitations:**

- the limitations are appropriately addressed in the supplementary material

---

> ### Author Rebuttal · Authors · 2023-08-09
>
> Thank you for your time and effort in reading our submission and writing your review!
>
> >**Line 94:** Sorry about this! Actually, every point is a small blue plus sign! We will explain this clearly in the caption.
>
> >**Q1:** We follow the evaluation strategy used to evaluate LiRA, where each model is trained on a different 50% split of the dataset. That ensures these models differ in their membership inference behavior.
>
> >**Q2:** The LiRA strategy trains many shadow models on a random 50% subsample of the teacher set. As a result, each teacher example will be present in roughly half of the shadow models, and will not be present in the other half. Then if we train 100 shadow models, we can look at their predictions, on each student example, of the ~50 shadow models containing the teacher, as well as the predictions of the ~50 shadow models without the teacher. We can do this for each student query, because we assume here the adversary sees the student query outputs (but cannot query on arbitrary points, which would be necessary to observe the model outputs for the teacher examples). This attack has the additional assumption that the adversary knows the student queries, as all of our End-to-End attacks do, but otherwise has no extra assumptions beyond LiRA’s. Let us know if this explanation helps, and we can add it to the paper, or help clarify more.

---

> > ### Comment · Reviewer_xTVE · 2023-08-13
> > **Answer Rebuttal**
> >
> > Thank you for your rebuttal.
> > All my questions were appropriately addressed, and I will raise my score from 7 --> 8 to `Strong Accept`.

---

### Official Review · Reviewer_E9Bb · 2023-07-06

**Soundness:** 4 excellent
**Presentation:** 4 excellent
**Contribution:** 4 excellent
**Rating:** 8
**Confidence:** 4

**Summary:**

The paper examines the effectiveness of model distillation in protecting the privacy of training data. Through the use of membership inference attacks, the authors demonstrate that distillation alone provides limited privacy across various domains. The authors also suggest several design considerations for improving privacy in model distillation, such as deduplicating the teacher set and considering complementary techniques like differentially private training.

**Strengths:**

- Trendy topic
- Well-organized paper
- Extensive experiments

**Weaknesses:**

- Attack detail in Section 5 is not clear
- Additional discussion might be helpful

**Questions:**

This paper provides valuable insights by addressing the misconception that model distillation can effectively protect the privacy of training data. The paper is well-organized and easy to follow, and a thorough examination of potential influencing factors contributes to the strength of the research.

However, I have some concerns regarding the details of the attacks and the interpretation of results:

- Regarding the experimental setting for End-to-End LiRA, it would be helpful to clarify how the corresponding student dataset was chosen. This information would enhance the reproducibility and validity of the experiments.

- The paper mentions that attacks on the student model should not be more powerful than those on the teacher model, as implied by the data processing inequality. However, Figure 3 reveals that some samples remain more vulnerable after distillation. It would be beneficial to provide a possible explanation for this observation, as it appears to contradict the expected outcome.

- Section 5, which introduces a new attack that only utilizes the student query dataset, is difficult to understand. The authors claim that “for each teacher example $z^T_j$, we fit a Gaussian distribution to the logits of each student example $z^S_i$, when $z^T_j$ is either IN or OUT.” However, it is unclear how the authors establish a link between student examples and arbitrary teacher examples, and how the student examples can indicate the membership status of the teacher example. Further clarification and additional illustrations would greatly enhance understanding in this section.

Overall, this paper makes significant contributions and offers valuable insights. Addressing the aforementioned concerns would further strengthen the research and improve the clarity of the presented findings.


**Limitations:**

The authors have adequately claimed their limitations in Appendix B.

---

> ### Author Rebuttal · Authors · 2023-08-09
>
> Thank you for your time in reading the submission and writing the review!
>
> >**End-to-End LiRA:** We use the same student set as used to train the target student model. This is because distillation can be done on public, nonsensitive data.
>
> >**Figure 3 & Data Processing Inequality:** The data processing inequality bounds the information contained in the student model about the data’s membership on the teacher. Thus, this only bounds the performance of an optimal attack, which LiRA on the teacher model may not be. We also note that the x and y coordinate of each point in Figure 3 is the average over a sample of target models and so are noisy values that can randomly appear above the dotted line.
>
> >**Clarification for Section 5:** The LiRA strategy trains many shadow models on a random 50% subsample of the teacher set. As a result, each teacher example will be present in roughly half of the shadow models, and will not be present in the other half. Then if we train 100 shadow models, we can look at their predictions, on each student example, of the ~50 shadow models containing the target teacher example, as well as the predictions of the ~50 shadow models without the target teacher example. Let us know if this explanation helps, and we can add it to the paper, or help clarify more.

---

> > ### Comment · Reviewer_E9Bb · 2023-08-13
> > **Thank you for the clarification**
> >
> > Thank you for the clarification. My questions were appropriately addressed!

---

### Official Review · Reviewer_4Gun · 2023-07-06

**Soundness:** 2 fair
**Presentation:** 4 excellent
**Contribution:** 3 good
**Rating:** 6
**Confidence:** 4

**Summary:**

The paper explores the privacy implications of model distillation, a technique used to transfer knowledge from a teacher model to a student model. The authors investigate membership inference attacks on both the teacher and student training sets to evaluate the privacy provided by distillation.
The authors extend the LiRA attack to the distillation setting in two ways. The first attack calibrates the attack only on the teacher models and the second attack uses the whole distillation procedure. They find that distillation alone provides limited privacy protection. The attacks on distilled models succeed even though the distilled models have never seen the teacher's data directly.
Finally, they also highlight the importance of considering factors such as data duplication, teacher set poisoning, temperature scaling and distribution shifts when evaluating the privacy risks of distillation.

**Strengths:**

- The authors address an important problem. As machine learning capabilities grow and models are deployed directly to personal devices with limited computing capability, model compression algorithms such as distillation are often used. Understanding the privacy risk of these algorithms is therefore an increasingly important topic.
- The paper is very easy to read and well motivated. Particularly, the visualizations in figure 3 and 4 are very intuitive and help tell the story
- The paper considers many experimental evaluation settings (albeit limited detail about the concrete experiments)

**Weaknesses:**

- An obvious mitigation of membership inference attacks is Differential Privacy, it would be interesting to see how effective DP mitigates the attack as by Mireshghallah et al [2022].
- Limited detail of experimental setup. What is the utility of the models?
- Some conclusions seem not deeply thought through. E.g. Line 213, the authors claim that since there are examples for which vulnerability drops by only  single digit percentage points and since privacy is a worst case guarantee the authors conclude that distillation provides limited privacy. While I agree that distillation provides limited privacy, it is possible that least vulnerable points see the single digit drop in vulnerability whereas the most vulnerable points see a more significant drop.


**References**

Mireshghallah, Fatemehsadat, et al. "Differentially private model compression." Advances in Neural Information Processing Systems 35 (2022): 29468-29483.

**Questions:**

What is the utility of the models? Is privacy simply correlated with utility? Do we see the same percentage point drops in model utility when going from teacher to student?

**Limitations:**

The authors have accurately described limitations albeit only in the appendix.

---

> ### Author Rebuttal · Authors · 2023-08-09
>
> Thank you for your time in reading the submission and writing the review!
>
> >**Differential privacy:** This is an interesting question. Part of the goal of our work was to consider distillation *without privacy* because there exist past works that attempt to show distillation can achieve a strong notion of privacy. Thus, our aim was to show this is not the case. Investigating differential privacy’s impact on these attacks is interesting future work.  We also remark that the Mireshghallah et al work considers a slightly different threat model where the teacher and student datasets are identical, which we show in the Supplemental material to be much more vulnerable to attack than the threat model where the teacher is sensitive and the student is public, nonsensitive data. We can also be more clear when we cite the Mireshghallah et al work in Section 2.2 that it would offer a provable guarantee against our attacks.
>
> >**Model utility:** Our CIFAR-10 models reach roughly 88% accuracy. This training setup reaches >94% accuracy on the full CIFAR-10 training set. On Purchase100 the student models reach 74-75% accuracy, and on Texas100 the models reach 54-55% accuracy (note these are 100 class tasks). Unfortunately, we didn’t save test predictions on WikiText, so we need to retrain a model to get its utility, please bear with us. We’ll put these numbers in the paper.
>
> >**Privacy as a worst case:** It’s true that the scenario you propose could offer worst-case privacy protection, but only if this were true for *all very vulnerable points*; we broadly find that this is not the case in Figure 3. We will be more careful about our wording here and ensure we highlight that not all vulnerable data see significant reductions in membership inference vulnerability.
>
> >**Privacy and utility?:** Though utility and vulnerability are often correlated, out attacks significantly outperform what can be inferred by simple attacks based on the accuracy of the model (i.e., the Yeom et al. 2018 attack, see Figure 12). Further, our attacks are not always well correlated with utility: there exist other factors that influence its success. For example, we find that increasing temperature (Figure 5d) does not change utility significantly, but does increase vulnerability. Distilling with CIFAR-100 examples (Figure 6) leads to less utility compared to distilling with CIFAR-10 examples (roughly 85% compared to 88%), but interestingly leads to higher vulnerability. Deduplicating increases utility slightly (88% instead of 87.5% accuracy), but reduces vulnerability (Figure 5b). We can add a discussion of this to the paper.

---

> > ### Comment · Reviewer_4Gun · 2023-08-12
> >
> > Thank you for the clarifications. Particularly for reporting the accuracy numbers. I think including them in the final version will help the reader the get a better picture of the experimental setting and help reproducibility.

---

### Official Review · Reviewer_odTu · 2023-07-09

**Soundness:** 4 excellent
**Presentation:** 4 excellent
**Contribution:** 3 good
**Rating:** 8
**Confidence:** 4

**Summary:**

The authors in this paper investigate the efficacy of membership inference attacks (MIA) in model distillation. Their novel attack(s) show that MIA is possible even when the teacher model is only queried on the most influential points in the student inputs. Finally, they also demonstrate how their attacks are the strongest when the teacher set is poisoned OR when the student set ~= teacher set.

**Strengths:**

1. The novelty is strong in this paper. No one has previously tackled MIA in model distillation as the authors here do.
2. The paper is also very easy to read and understand.
3. I really like the novel idea present in this paper (specifically Figure 1): membership of a target example in the teacher’s training set can be derived through querying the teacher model on various student set examples (and these are entirely different examples from the target!).
4. Figure 3 is a great result as well. It might not be as obvious as it is, because the data points are essentially “post-processed” and there should be some sense of privacy after the distillation process. The authors here, show that distillation doesn’t show much privacy benefits.
5. Their investigation on why MIA works well for student models is sound. I really like Figure 5 as well, which shows the MIA effects through duplicates, data poisoning and temperature scaling and the section on student-teacher data drift was an amazing read :-)

**Weaknesses:**

1. I don’t really have any major weaknesses to point out!

**Questions:**

1. Could this happen because of correlated features present between the student queries and the target example? Acc to Figure 1, the “red” colour present in the student examples are essential for the MIA to work. My intuition is that a higher correlation in features between the student inputs and the teacher target example, should lead to higher success in MIA. Of course, this may be written down as Future Work and I’m not quite sure if such work has been done in this space. I think it might be nice to investigate such correlation effects between the student and teacher examples.
2. I noticed that all the attacks mentioned in this paper, were based off LiRA. Is there a specific reason why the authors succumbed to using LiRA and not other attacks. (It's totally okay to use LiRA but I wonder if there could be other attacks you could have tried/did try out and measure the efficacy of such attacks against the LiRA-based ones; provided you have time to run these expts).

**Limitations:**

1. Need to run a number of shadow models for running a successful MIA (not really a limitation of this approach, but this is a limitation of MIA itself!).

---

> ### Author Rebuttal · Authors · 2023-08-09
>
> Thank you for your time in reading our submission and interesting questions!
>
> >**Feature correlation:** This is an interesting intuition, and we agree that the example in Figure 1 does seem to be due to this “red”ness (and one might draw similar conclusions about the examples in Figure 9 in the supplement). However, it is unclear how to define and quantify correlation for systematic study: we believe this is very interesting future work. This seems to have implications beyond privacy in understanding distillation’s dark knowledge, and we were unable to find related results in this literature.
>
> >**Why use LiRA:** LiRA represents the current state-of-the-art for membership inference. Thus, it represents the strongest baseline and also the strongest starting point for our research. Prior work using distillation as a defense has all predated the LiRA paper and so only evaluated with weaker attacks. This is also a motivation for our work: to understand whether stronger membership inference attacks could challenge this use of distillation. We also see this in Figure 12 in the supplementary material: a simple logit-gap membership inference attack (akin to simpler attacks such as Yeom et al. 2018) is unable to get beyond random guessing on CIFAR-10.

---

> > ### Comment · Reviewer_odTu · 2023-08-11
> > **Addressing the rebuttal #1**
> >
> > Thank you for your rebuttal!
> >
> > I am happy with the responses and I have increased my score by another point (7 --> 8).
> >
> > On a similar note, I found this paper on `déjà vu memorization` (https://arxiv.org/abs/2304.13850) which might be relevant for quantifying memorization through feature correlations. For future work, it might be nice to explore threat models around how privacy could be compromised by learning sensitive (correlated) features!

---

### Decision · Program_Chairs · 2023-09-21

**Decision:**

Accept (oral)

**Comment:**

The paper discusses a crucial issue of privacy in the face of model distillation.  Reviewers unanimously agree on the significance of the contributions made. Strengths highlighted by the reviewers include the paper's remarkable clarity and the thoroughness of experiments.

The reviewers also highlight the paper's contribution by dispelling a misconception that distillation safeguards data privacy. By spotlighting factors such as data duplication, teacher set poisoning, temperature scaling, and distribution shifts, the authors enrich the discourse on the topic.